# Structure of a mitochondrial ATP synthase with bound native cardiolipin

Alexander Mühleip[1,2], Sarah E McComas[1], Alexey Amunts[1,2]*

[1]Science for Life Laboratory, Department of Biochemistry and Biophysics, Stockholm University, Solna, Sweden; [2]Department of Medical Biochemistry and Biophysics, Karolinska Institutet, Stockholm, Sweden

**Abstract** The mitochondrial ATP synthase fuels eukaryotic cells with chemical energy. Here we report the cryo-EM structure of a divergent ATP synthase dimer from mitochondria of *Euglena gracilis*, a member of the phylum Euglenozoa that also includes human parasites. It features 29 different subunits, 8 of which are newly identified. The membrane region was determined to 2.8 Å resolution, enabling the identification of 37 associated lipids, including 25 cardiolipins, which provides insight into protein-lipid interactions and their functional roles. The rotor-stator interface comprises four membrane-embedded horizontal helices, including a distinct subunit *a*. The dimer interface is formed entirely by phylum-specific components, and a peripherally associated subcomplex contributes to the membrane curvature. The central and peripheral stalks directly interact with each other. Last, the ATPase inhibitory factor 1 (IF$_1$) binds in a mode that is different from human, but conserved in Trypanosomatids.

## Introduction

The mitochondrial ATP synthase is a membrane protein complex that generates most of the ATP in eukaryotic cells. The synthesis of ATP from ADP and inorganic phosphate proceeds via rotary catalysis, which uses the energy of the electrochemical gradient across the mitochondrial inner membrane. The translocation of protons through the membrane-bound F$_o$ part, mediated by subunit *a*, drives the rotation of a membrane-embedded c-ring and the attached central stalk, which together form the rotor. The torque of the rotor against the stator subunits induces conformational changes in the (αβ)$_3$ headpiece, thereby triggering catalysis (*Abrahams et al., 1994*; *Noji et al., 1997*). The mitochondrial ATP synthase forms dimers, which in turn associate into dimer rows along the high-curvature membrane regions of the cristae (*Davies et al., 2012*; *Paumard et al., 2002*; *Strauss et al., 2008*). Loss of ATP synthase dimers results in aberrant cristae morphology, indicating that dimers are required for membrane bending and proper cristae formation in mitochondria (*Davies et al., 2012*; *Paumard et al., 2002*).

Previous biochemical and mass spectrometry analysis showed that a highly divergent ATP synthase with numerous new subunits is found in the phylum of Euglenozoa, belonging to the Excavata supergroup (*Perez et al., 2014*; *Yadav et al., 2017*; *Zíková et al., 2009*). A combination of transcriptome analysis and sequencing of nuclear and mitochondrial genomes showed the phylogenetic relationship between the protozoan *Euglena gracilis* and Kinetoplastids, which include human parasites such as *Trypanosoma* and *Leishmania* that cause sleeping sickness, Chagas disease, and Leishmaniasis (*Dobáková et al., 2015*; *Ebenezer et al., 2019*). However, while the reported recruitment of additional subunits in the Euglenozoan ATP synthase implies a distinct architecture, neither the complete composition, nor the functions or structures of the specific subunits have been reported. Previous attempts were limited by resolution (*Mühleip et al., 2017*) or restricted to the conserved F$_1$ subcomplex (*Montgomery et al., 2018*). In addition, lipids are known to be important to the structure and function of mitochondrial ATP synthases (*Eble et al., 1990*; *Kühlbrandt, 2019*;

*For correspondence:
amunts@scilifelab.se

Competing interests: The authors declare that no competing interests exist.

**eLife digest** Every living thing uses the energy-rich molecule called adenosine triphosphate, or ATP, as fuel. It is the universal molecular currency for transferring energy. Cells trade it, mitochondria make it, and the energy extracted from it is used to drive chemical reactions, transport molecules across cell membranes, energize nerve impulses and contract muscles.

ATP synthase is the enzyme that makes ATP molecules. It is a multi-part complex that straddles the inner membrane of mitochondria, the energy factories in cells. The enzyme complex interacts with fatty molecules in the mitochondrial inner membrane, creating a curvature that is required to produce ATP more efficiently. The mitochondrial ATP synthase has been studied in many different organisms, including yeast, algae, plants, pigs, cows and humans. These studies show that most of these ATP synthases are similar to each other, but obtaining a high resolution structure has been a challenge.

Some single-cell organisms have unusual ATP synthases, which provide clues about how the enzyme evolved in pursuit of the most energy efficient arrangement. One such organism is the photosynthetic *Euglena gracilis*, which is closely related to the human parasites that cause sleeping sickness and Chagas disease.

Now, Mühleip et al. have extracted ATP synthase from *E. gracilis* and reconstructed its structure using electron cryo-microscopy. The high resolution of this reconstruction allowed for the first time to examine the fatty molecules associated with ATP synthase, called cardiolipins. This is important, because cardiolipins are thought to modulate the rotating motor of the enzyme and affect how the complex sits in the membrane.

The analysis revealed that the ATP synthase in *E. gracilis* has 29 different protein subunits, 13 of which are only found in organisms of the same family. Some of the newly discovered subunits are glued together by fatty molecules and extend into the surrounding mitochondrial membrane. This distinctive structure suggests an adaptation which likely evolved independently in *E. gracilis* for efficiency.

These results represent an important advance in the field, and provide direct evidence for the functional roles of cardiolipin. This information will be used to reconstruct the evolution of this mighty molecule and to further study the roles of cardiolipin in energy conversion. Moreover, the analysis identified similarities between the ATP synthase in *E. gracilis* and human parasites, which could provide new therapeutic targets in disease-causing parasites.

*Laage et al., 2015*; *Srivastava et al., 2018*). Proper cristae formation specifically depends on cardiolipin (CL) (*Mileykovskaya and Dowhan, 2009*), an anionic lipid that accounts for 20% of total lipids in the inner mitochondrial membrane (*Calvayrac and Douce, 1970*; *Daum and Vance, 1997*; *Zinser and Daum, 1995*). Cardiolipin is also essential for the activity of isolated mitochondrial ATP synthases (*Laird et al., 1986*; *Pitotti et al., 1972*; *Santiago et al., 1973*), and promotes the formation of dimer rows (*Acehan et al., 2011*). Although bound lipids have previously been observed in structures of rotary ATPases (*Klusch et al., 2017*; *Murphy et al., 2019*; *Vasanthakumar et al., 2019*), the atomic models of recently reported cryo-EM structures of mitochondrial ATP synthases do not include cardiolipin (*Gu et al., 2019*; *Guo et al., 2017*; *Klusch et al., 2017*; *Murphy et al., 2019*; *Srivastava et al., 2018*).

To reveal how the mitochondrial ATP synthase is modulated by lipids and organised into a dimer shaping discoid cristae in Euglenozoan mitochondria, we determined the structure of the ATP synthase dimer from *E. gracilis*. The atomic model of the entire 2-MDa complex contains 29 different subunits (eight newly identified) and provides a comprehensive description of a markedly distinct mitochondrial ATP synthase dimer, including a previously unseen binding mode of its natural inhibitor protein $IF_1$. The membrane region was determined to 2.8 Å resolution, enabling the identification of a structurally divergent subunit *a* and visualisation of 37 associated native lipids. Importantly, cardiolipin binding sites are found at the rotor-stator interface, dimer interface, and in a peripheral $F_o$ cavity. These data provide insight into protein-lipid interactions in the mitochondrial ATP synthase and its evolution, suggesting functional roles of lipids in proton translocation, dimerization and stabilisation.

## Results and discussion

### Overall structure

The mitochondrial ATP synthase dimer was purified natively from *E. gracilis* and subjected to cryo-EM structure determination (*Supplementary file 1*). Symmetry expansion of the pseudo-$C_2$-symmetric dimer particles was used to classify individual $F_1$/c-ring monomers into rotational states 1, 2 and 3 (named according to bovine nomenclature; *Zhou et al., 2015*) and resolve structures at 3.0- to 3.9 Å resolution. Using masked refinement, maps of the membrane region, the rotor and the peripheral stalk tip were refined to 2.8 and 3.3 and 3.8 Å resolution, respectively, enabling the construction of atomic models (*Figure 1—figure supplements 1–2*; *Video 1*; *Supplementary file 2*). The model of the complete mitochondrial ATP synthase contains 29 different proteins, of which 14 are phylum-specific, displaying a distinct architecture, particularly in the membrane region. The interactions between the monomers are expanded, resulting in a 45° dimer angle, compared to ~100° in yeast and mammals (*Figure 1A*) (*Davies et al., 2012*; *Gu et al., 2019*; *Hahn et al., 2016*). The membrane-bound $F_o$ region is composed of 22 proteins, of which 13 have no homologs in animals and fungi. The well-resolved membrane region allowed the identification of 37 native lipids, leading to the assignment of 25 cardiolipins, whereas the 12 remaining phophoslipids could not be unambiguously identified and were modelled as phosphatidic acid.

Following reports of mitochondrial ATP synthases from various organisms, different names have previously been given to subunits performing the same role in F-type ATP synthases (*Kühlbrandt, 2019*). For the mitochondrial ATP synthase from *E. gracilis*, we adopt a nomenclature that is consistent with the conserved $F_o$ subunits from yeast (subunits *a-d*, *f*, *i/j*, *k*, 8, OSCP) (*Guo et al., 2017*) and euglenozoa-specific subunits previously identified in *Trypanosoma brucei* (ATPTB1, 3, 4, 6, 12) (*Perez et al., 2014*; *Zíková et al., 2009*), whereas the additional subunits are named ATPEG1 to 8, according to their molecular weight (*Supplementary file 3*).

### Identification of the transmembrane subunits

All known mitochondrial ATP synthases form dimers in the membrane through transmembrane $F_o$ subunits. However, in *E. gracilis*, only subunit *c* and a set of phylum-specific subunits were identified (*Perez et al., 2014*; *Yadav et al., 2017*; *Zíková et al., 2009*), suggesting a divergent $F_o$ composition. The core subunit *a* mediates proton translocation and contains the strictly conserved R176 (*Saccharomyces cerevisiae* numbering). Despite its functional importance, subunit *a* was identified neither in the *E. gracilis* genome project (*Ebenezer et al., 2019*), nor in the mitochondrial genome and transcriptome analysis (*Dobáková et al., 2015*). Our cryo-EM map allowed tracing of subunit *a*, for which we identified six structurally conserved membrane-embedded helices (H1$_a$ to H6$_a$) directly from side-chain densities (*Figure 2—figure supplement 1A* to C). Using this information, we then found the matching sequence in the available genomic data and mapped it to a mitochondrial contig, which also contains subunit 8 and *nad1* in a single open reading frame (*Figure 2—figure supplement 1E*). Thus, by combining the information from cryo-EM and sequencing, we report the most divergent subunit *a* found up to date.

In all previously reported ATP synthase structures, the horizontal helix H5$_a$ bends around the c-ring, following its curvature, thereby contacting four of the ten subunits in the c-rings of yeast and algae (*Figure 2B,D*) (*Allegretti et al., 2015*; *Guo et al., 2017*; *Hahn et al., 2016*). By contrast, in the *E. gracilis* structure, the N-terminal part of H5$_a$ is kinked towards the lumen, and therefore does not interact with the c-ring, and instead extends towards the lumenal membrane surface (*Figure 2A,C*). This structural rearrangement results in a smaller interface between subunits *a* and *c*, with only three c-subunits forming

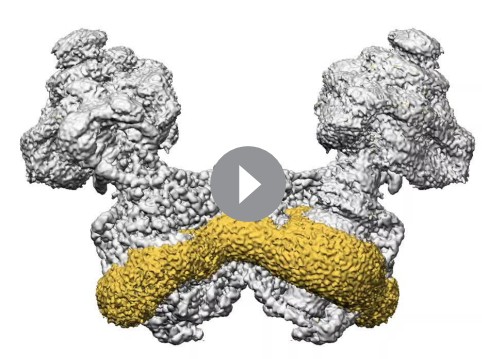

**Video 1.** Density map of *E. gracilis* ATP synthase dimer with regions corresponding to protein shown in grey and the detergent belt coloured gold.
https://elifesciences.org/articles/51179#video1

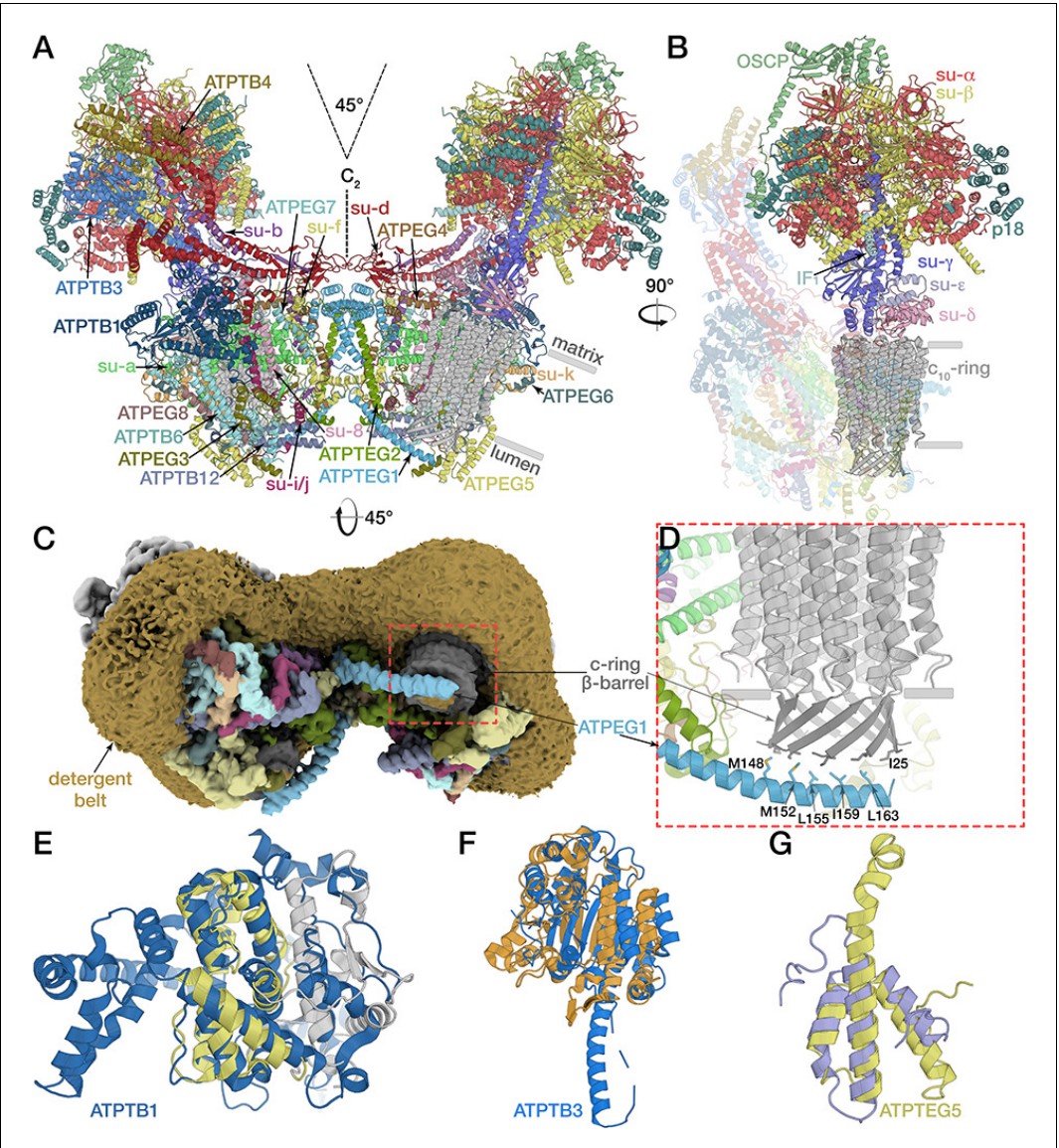

**Figure 1.** Structure of the *E. gracilis* ATP synthase dimer. (**A**) Atomic model of the complete *E. gracilis* ATP synthase dimer with both subcomplexes in rotational state-1. The 2-MDa dimeric $F_1F_o$-complex contains 29 different proteins. Dashed lines indicate $C_2$-symmetry axis and 45˚ dimer angle. (**B**) OSCP/$F_1$/c-ring subcomplex in rotational state-1, bound to its natural inhibitor protein $IF_1$ (cyan), remaining $F_o$ transparent. (**C**) Density map showing the lumen-exposed $F_o$ region. Detergent belt shown in yellow; c-ring β-barrel in dark grey, $F_o$ subunits as in (**A**). (**D**) Close-up of the lumenal interface of ATPEG1 (blue) with the c-ring (grey). The interaction occurs mostly via hydrophobic interactions (blue and grey sticks). (**E–G**) Euglenozoa-specific $F_o$-subunits with known folds. (**E**) ATPTB1 in blue superposed with Mdm38 (PDB ID: 3SKQ) (*Lupo et al., 2011*), six conserved helices coloured yellow, rest grey. (**F**) ATPTB3 in blue superposed with a bacterial homoisocitrate dehydrogenase in orange (PDB ID: 4YB4)(*Takahashi et al., 2016*), adopts a Rossman-fold. (**G**) ATPEG5 in yellow is a structurally conserved ortholog of the cytochrome c oxidase subunit VIb superfamily; bovine subunit VIb in purple (PDB ID: 2Y69) (*Kaila et al., 2011*).

The online version of this article includes the following figure supplement(s) for figure 1:

**Figure supplement 1.** Cryo-EM data processing and classification scheme.
**Figure supplement 2.** Local resolution estimation and model-map-correlations.

interactions, which also has implications for the formation of the matrix half-channel, as discussed below.

In the yeast mitochondrial ATP synthase, subunit *a* interacts with transmembrane subunits *b*, *f*, *i*/*j*, *k*, 8 and membrane-associated subunit *d* (*Guo et al., 2017*). Since no homologs were reported for any of these subunits in *E. gracilis* or *T. brucei* (*Perez et al., 2014*; *Yadav et al., 2017*; *Zíková et al., 2009*), we next investigated their putative location through superimposition of our F$_o$ model with the yeast counterpart (*Guo et al., 2017*). Based on the matching position and topology of the transmembrane helices as well as conserved positions of interactions with subunit *a*, we identified all six associated subunits, which are structurally conserved, but display no significant sequence similarity to yeast counterparts (*Figure 3*). Subunits *b*, *f*, *i*/*j*, *k* and 8 contain a single transmembrane

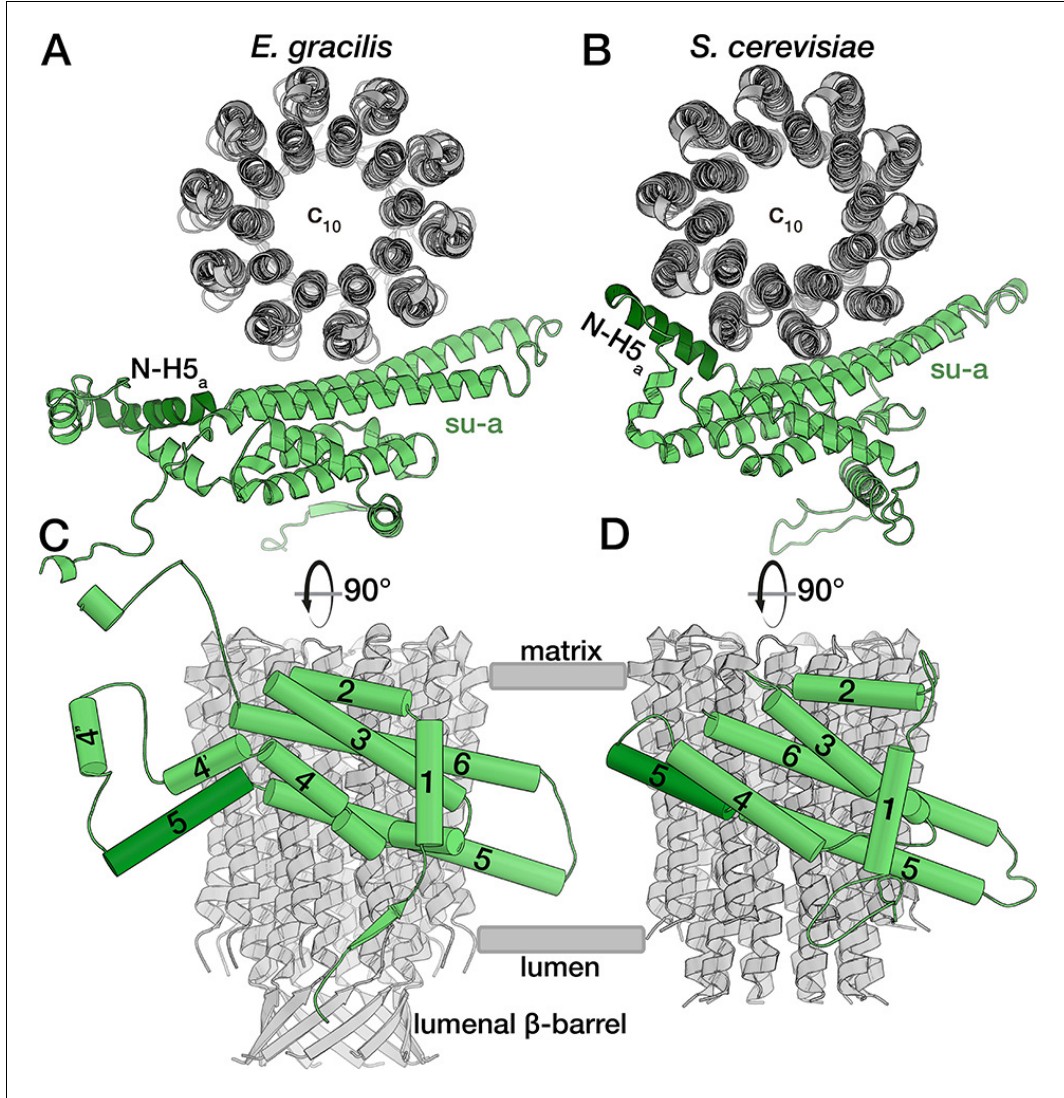

**Figure 2.** *E. gracilis* subunit *a* fold comparison. Top view (upper panel) and side view (lower panel) of the *E. gracilis* (left) and *S. cerevisiae* (right) (*Guo et al., 2017*) subunit *a* (green) and c-ring (grey). Both structures contain the conserved H1-6$_a$, with *E. gracilis* displaying two helices (H4'$_a$ and H4''$_a$) in an extension segment and a C-terminal extension. Whereas the N-terminal region of H5$_a$ (dark green) is kinked towards the c-ring in the yeast complex, it extends towards the lumen in the *E. gracilis* structure, thereby diminishing its interface with the c-ring. Unlike its yeast homolog, the N-terminus of *E. gracilis* subunit *a* is not involved in dimerisation, but contributes a strand to a β-sheet along the lumenal side of the detergent micelle.

The online version of this article includes the following figure supplement(s) for figure 2:

**Figure supplement 1.** Identification of subunits *a* and 8.

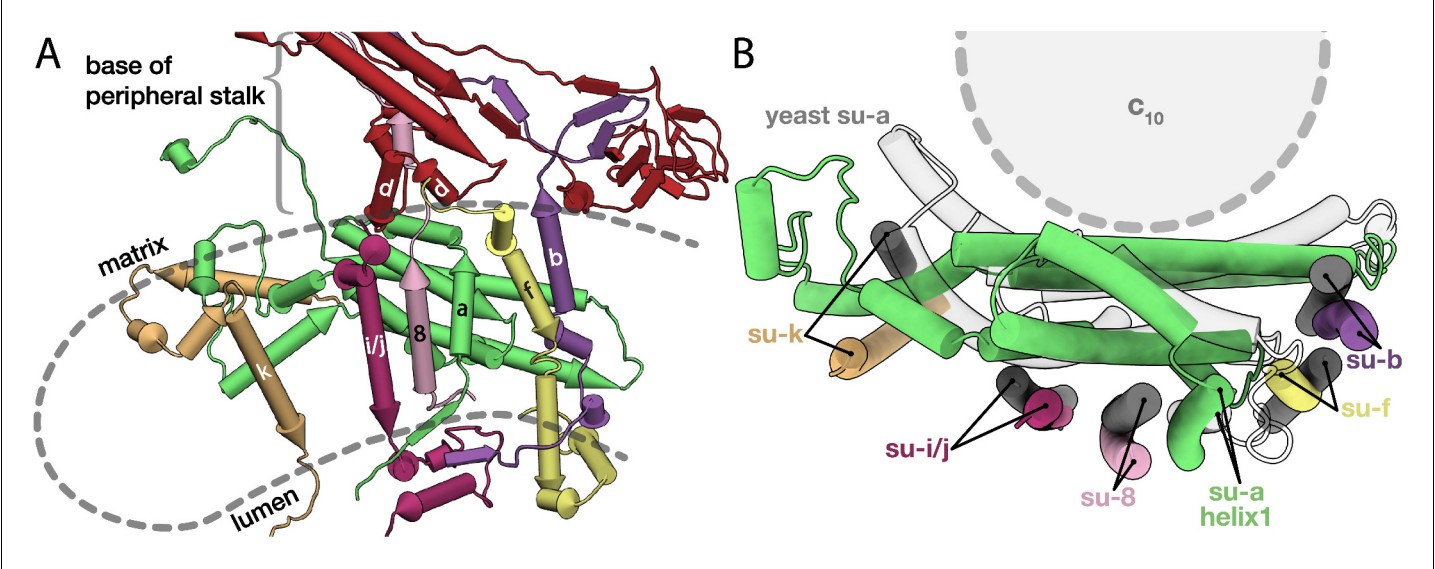

**Figure 3.** Conserved subunits of the $F_o$ region. (**A**) Side view of the conserved *E. gracilis* $F_o$ subunits. Transmembrane helices with structural equivalents in yeast are labelled. (**B**) Top view of the superimposed conserved $F_o$ subcomplexes from *E. gracilis* (coloured) and yeast (grey) PDB ID: 6B2Z (*Guo et al., 2017*). Although subunit *k* does not superimpose well, it occupies the same position relative to the H5$_a$.

helix associated with subunit *a*, whereas subunit *d* forms a clamp around subunit 8 at the base of the peripheral stalk, containing a structurally conserved two-helix motif at the matrix side of the membrane (*Figure 3A*). Finally, subunit *k* is bound peripherally to subunit *a*, as in yeast, however the H5$_a$ kink results in a 15-Å displacement of the transmembrane helix of subunit *k* away from the c-ring, compared to its yeast counterpart (*Figure 3B*). These data show that despite sequence divergence, the assembly of the central $F_o$ subunits around subunit *a* is architecturally conserved between Euglenozoa and Metazoa.

The striking architectural divergence of the *E. gracilis* ATP synthase dimer is brought about by euglenozoa-specific subunits and extensions of the structurally conserved $F_o$ subunits, which render them on average 2.5 times larger than in the yeast mitochondrial ATP synthase. Only subunit *b* is truncated. The extensions of the conserved $F_o$ subunits are mostly involved in forming interactions with the euglenozoa-specific subunits, thus providing a platform for the observed increased molecular mass of the $F_o$ (*Figure 4—figure supplement 1A,C*).

The additional 13 euglenozoa-specific $F_o$ subunits determine the architecture of the ATP synthase dimer, giving the $F_o$ a markedly different overall shape, making it almost three times the size of its yeast counterpart (*Figure 4—figure supplement 1*). They contribute to the dimerization interface, the peripheral stalk and $F_o$ periphery. The C-terminal helix of euglenozoa-specific ATPEG1 (H5$_{EG1}$) extends 50 Å from the membrane region into the lumen, where it interacts with the N-terminal extensions of subunit *c*, which together form a ten-stranded β-barrel (*Figure 1C and D*) protruding ~20 Å into the lumen. The N-terminal residues of subunit *c* (A24, I25) form a hydrophobic interface with hydrophobic residues (M148, M152, L155, I159, L163) of the amphipathic ATPEG1 helix. The position of the lumenal H5$_{EG1}$ on the c-ring β-barrel remains largely unchanged in all three rotational states, suggesting a mechanism of transient rotor-stator interaction during c-ring rotation. A similar lumenal interaction has previously been reported in the bovine ATP synthase, where subunit *e* extends from the membrane to contact the c-ring (*Zhou et al., 2015*). In the porcine ATP synthase tetramer, subunit *e* has been proposed to interact with the 6.8 kDa proteolipid, which has been suggested to reside inside the c-ring (*Gu et al., 2019*).

Other euglenozoa-specific $F_o$ subunits contain structural domains that were shown to be functionally important in mitochondria (*Figure 1E* to G). ATPTB1 is a membrane-associated protein on the matrix side of the $F_o$ periphery that adopts an Mdm38-like fold, which was shown to associate with yeast mitochondrial ribosomes at the inner mitochondrial membrane (*Frazier et al., 2006*). ATPTB3 is an isocitrate dehydrogenase ortholog that adopts a Rossman fold located at the tip of the

peripheral stalk. ATPEG5 is a structurally conserved ortholog of the cytochrome c oxidase subunit VIb superfamily.

## Dimer interface and associated lipids

The defining feature of mitochondrial ATP synthases is the formation of dimers in the crista membrane. In yeast, the two monomers are connected through $F_o$, and the dimer interface is formed on the lumenal side by the conserved $F_o$ subunits $a$ and $i/j$, as well as subunits $k$ and $e$ (*Guo et al., 2017*). Our atomic model of the *E. gracilis* mitochondrial ATP synthase shows that in contrast to yeast, subunits $a$, $i/j$, $k$ do not contribute to the dimer interface, which instead is formed by species- and phylum-specific subunits and extensions of apparent homologs (*Figure 4*, *Figure 4—figure supplement 1*, *Figure 4—figure supplement 2*). The extensive dimer contacts are stacked across three layers: the matrix side, the transmembrane region and the lumenal side (*Figure 4A* and *Figure 4—figure supplement 1A,C*). On the matrix side, an extension of subunit $d$ adopts an elaborated ferredoxin-like all-β fold that forms a dimer interface close to the symmetry axis (*Figure 4A* and *Figure 4—figure supplement 1E,F*). The curved H3 of ATPEG1 forms a homotypic dimerisation motif along the matrix surface of the membrane plane. ATPEG2 contributes to the dimer interface with both a transmembrane helix that interacts with ATPEG1 and its termini, which link the two $F_o$-parts, each extending into the rotor-stator interface of one monomer. On the lumenal surface, ATPEG1 interacts with the C-terminal extension of subunit $f$. The different $F_o$ subunit composition and their involvement in dimer formation result in a 45° dimer angle, compared to ~100° in yeast (*Figure 1A*) (*Guo et al., 2017*). Thus, despite the presence of conserved $F_o$ subunits and the contribution to their extensions to the dimer interface (*Figure 4—figure supplement 2*), the *E. gracilis* mitochondrial ATP synthase displays a fundamentally different dimer architecture, when compared to the yeast (*Figure 4—figure supplement 1*), mammalian and *Polytomella* ATP synthases (*Gu et al., 2019*; *Murphy et al., 2019*), suggesting that dimer formation evolved independently in different lineages.

In addition to the described protein-protein interactions, we identified nine bound phospholipids occupying the dimer interface (*Figure 4C* to G, *Figure 4—figure supplement 2*). Five of them are cardiolipin molecules linking dimerising subunits close to the $C_2$-symmetry axis (*Figure 4C* to E). CDL11 links two horizontal helices of the two symmetry-related copies of ATPEG1, which extend along the matrix side of the membrane region (*Figure 4E*, *Video 2*). These protein-lipid interactions indicate a functionale role of cardiolipin in the stabilisation of the dimer contacts, which is consistent with its proposed role in mediating subunit interactions between the transmembrane helices in mitochondrial supercomplexes (*Mileykovskaya and Dowhan, 2014*; *Wu et al., 2016*).

## Rotor-stator interface and proton path

Proton translocation occurs at the rotor-stator-interface, which is canonically formed in the membrane by horizontal helices $H5_a$, $H6_a$ and the c-ring (*Allegretti et al., 2015*) (*Figure 5—figure supplement 1B,D*). In the *E. gracilis* ATP synthase, the essential R178 of $H5_a$ is conserved, and interacts with Gln232 on $H6_a$. Intriguingly, we found two additional horizontal helices, contributed by ATPEG4 ($H1_{EG4}$) and subunit $k$ ($H1_k$) (*Figure 5A,C* and *Figure 5—figure supplement 1A*). ATPEG4 is bound to subunit $a$ in the membrane, and its amphipathic, horizontal helix ($H1_{EG4}$) is positioned close to the matrix side of the membrane, extending in parallel to $H6_a$ at a distance of ~13 Å (*Figure 5A*), allowing it to interact with the same transmembrane helix of the c-ring as $H5_a$ and $H6_a$ (*Figure 5—figure supplement 2A*).

The *E. gracilis* ATP synthase structure suggests a mechanism of proton translocation via two offset proton half channels, similar to those previously described in other F-type ATP synthases (*Allegretti et al., 2015*; *Guo et al., 2017*; *Hahn et al., 2016*). Protons enter the membrane region via the lumenal half-channel, which we traced as an internal cavity of the atomic model. The entrance of the lumenal half-channel is lined by the C-termini of subunit $f$ and ATPEG4, extension of subunit $i/j$ and the N-terminal regions of ATPEG3 and subunit 8, of which only the latter is structurally conserved in yeast (*Figure 5B*). Inside the $F_o$ region, the lumenal half-channel is lined by the conserved transmembrane helices of subunits $f$ and $b$ and extends between $H5_a$, $H6_a$, as previously suggested (*Guo et al., 2017*). At the exit of the lumenal half-channel, we identified a β-DDM detergent molecule, with its hydrophilic maltose head group protruding into the channel and the acyl chain

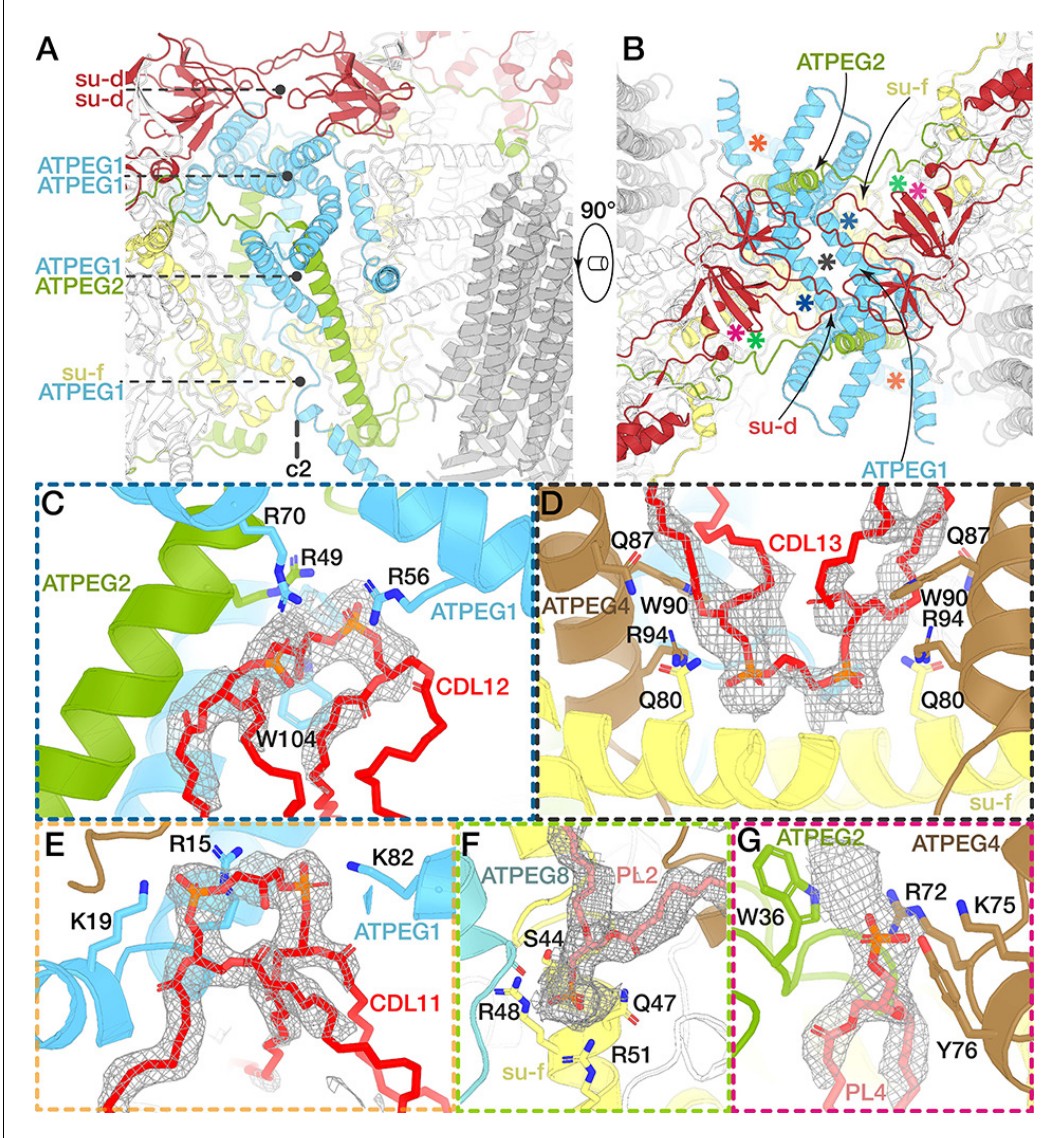

**Figure 4.** The dimer interface and associated lipids. (**A and B**) Views of the dimer interface along (**A**) and perpendicular (**B**) to the membrane plane. The dimerisation motifs (interacting subunits coloured) are stacked along the $C_2$-symmetry axis and formed by two copies of subunit *d* (red) and ATPEG1 (blue), which interacts with its symmetry-related copy, as well as ATPEG2 (green) and subunit *f* (yellow). Asterisks in (**B**) indicate positions of lipid-binding sites. (**C to G**) Close-ups of the lipid-binding sites indicated in (**B**). Bound lipids at the dimer interface identified as cardiolipin (CDL; C to E) or phospholipids modelled as phosphatidic acid (PL; **F and G**). Interacting residues (subunits coloured) include at least one arginine residue. Density shown as grey mesh.

The online version of this article includes the following figure supplement(s) for figure 4:

**Figure supplement 1.** Architecture and dimer interface comparison between the yeast and *E. gracilis* ATP synthase.

**Figure supplement 2.** Species-specific subunits and extensions form the dimer interface.

**Figure supplement 3.** Bound lipids of the dimer interface.

extending between H5$_a$ and H6$_a$, confirming both the hydrophilic environment and accessibility of the membrane-intrinsic lumen channel (*Figure 5B*).

Proton translocation to the rotor-stator interface results in protonation of the conserved glutamate residue (E86 in *E. gracilis*) in the middle of the c-ring and subsequent counter-clockwise rotation of the c-ring when viewed from the F$_1$ to F$_o$ (*Noji et al., 1997*). This proton transfer has been suggested to be ultimately mediated by a glutamate of H6$_a$ (E223 in *S. cerevisiae*), which is paired

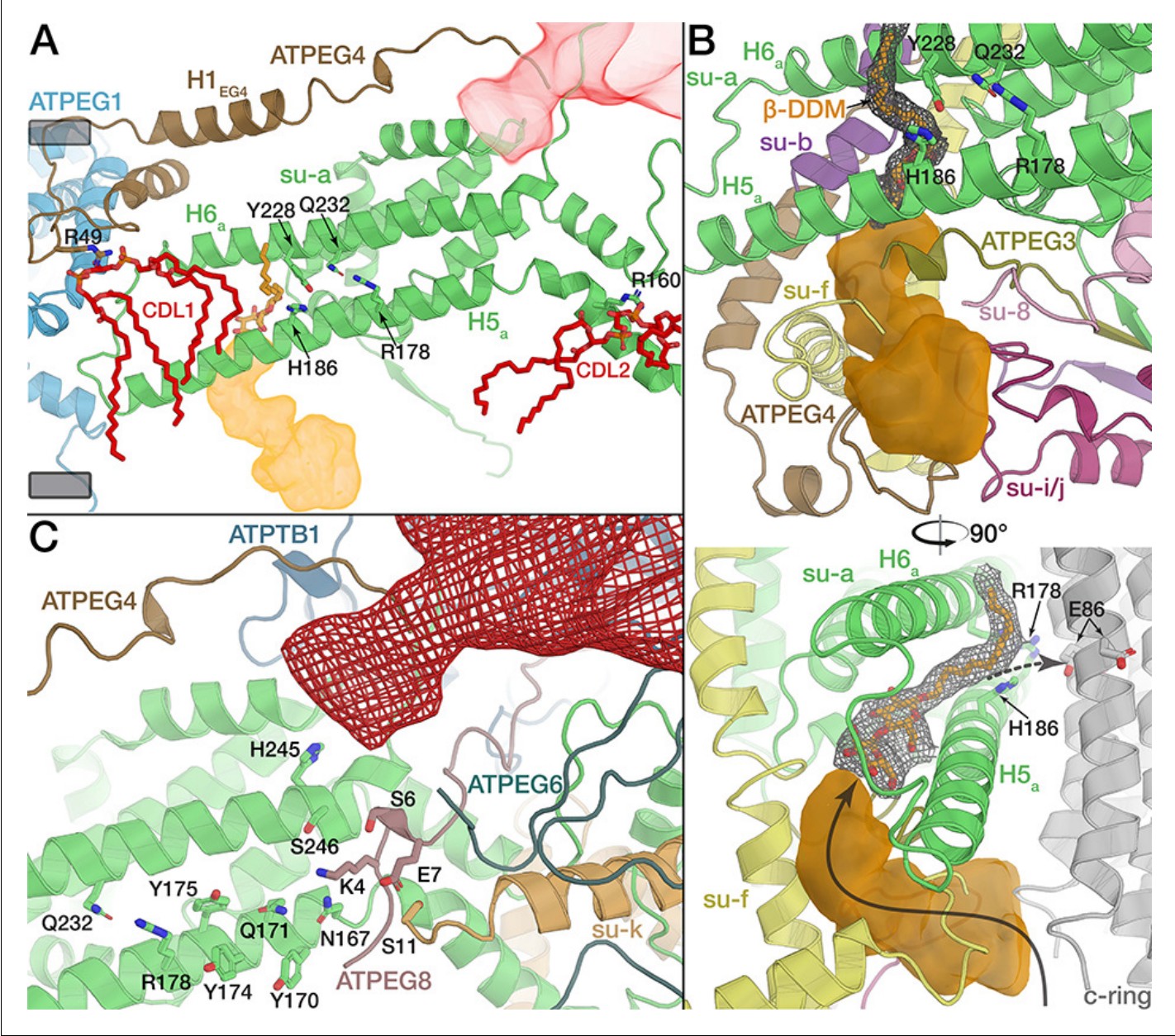

**Figure 5.** The rotor-stator interface is flanked by bound cardiolipin. (**A**) View from the c-ring towards the membrane-embedded stator subunits. H5$_a$ and H6$_a$ are augmented by the tilted, amphipathic H1$_{EG4}$ (brown). Cardiolipin molecules flanking subunit *a* are shown in red (tails of acyl chains are mostly disordered and shown only for illustration). Proton half-channels on the lumen and matrix side are shown in orange and red respectively. Remaining subunits not shown for clarity. The conserved R178 and the H186 at the lumen channel exit are shown with interacting residues. (**B**) Entrance of the lumenal channel (orange) is lined by the termini of subunit *f*, ATPEG4, subunit 8, ATPEG3, as well as a lumenal segment of subunit *i/j*. Inside the F$_o$, the lumen channel is confined by transmembrane helices of subunits *f* and *b*. β-DDM occupying the exit of the lumen channel shown in orange with density map as mesh, c-ring in grey. Arrows indicate proposed path of proton flow. (**C**) Polar and protonatable residues between R178 and the matrix-side half channel (red mesh). Subunit *k* contributes a horizontal helix (H1$_k$) to the rotor-stator interface.

The online version of this article includes the following figure supplement(s) for figure 5:

**Figure supplement 1.** Comparison between *E. gracilis* and *S. cerevisiae* rotor-stator interfaces.

**Figure supplement 2.** Native phospholipids of the rotor-stator interface.

---

with H185 of H6$_a$. This residue pair facing the lumenal half-channel is conserved in yeast, mammals and *Polytomella* (*Gu et al., 2019*; *Guo et al., 2017*; *Klusch et al., 2017*) but surprisingly not in *E. gracilis* (V225 and S187, respectively; *Figure 5A*). The absence of an acidic residue from the exit of

the lumenal proton half-channel indicates that it is not strictly required for proton transfer to the c-ring in mitochondrial ATP synthases. Instead, this function appears to be compensated in our structure by H186 of $H5_a$, which extends towards the c-ring and interacts with Y228 of $H6_a$, thus forming an alternative residue pair at the lumenal channel exit (*Figure 5B*).

After almost a full rotation of the c-ring, the glutamate residue (E86) is deprotonated by R178 of subunit *a*. The translocated proton is then released into the matrix half-channel. In other ATP synthases, $H5_a$ bends around the *c*-ring, thereby determining the channel path (*Guo et al., 2017*). Due to the unusually kinked $H5_a$, *E. gracilis* ATP synthase lacks the interaction of the N-terminal segment of $H5_a$ following the c-ring curvature (*Figure 2A*). Instead, its functional role in forming the release channel is replaced by the N-termini of ATPEG4, ATPEG6 and ATPEG8 and subunit *k* which forms a fourth horizontal helix in the membrane that extends towards the c-ring (*Figure 5C*, *Figure 5—figure supplement 2A*). Thus, the reduced interface between subunits *a* and *c* is compensated by species-specific structural elements forming the matrix half channel (*Figure 5C*). Taken together with the horizontal $H1_{EG4}$ and the lumenal $H5_{EG1}$, the *E. gracilis* ATP synthase displays and increased number of c-ring interactions compared to its yeast counterpart. As a consequence, the *E. gracilis* rotor-stator interface displays larger buried surface area of ~830 $Å^2$ compared to ~480 $Å^2$ in yeast (*Figure 5—figure supplement 1C and D*).

Adjacent to the proton-half channels, the *E. gracilis* ATP synthase structure reveals two bound cardiolipins (CDL1, CDL2) flanking either side of the two horizontal helices $H5_a$ and $H6_a$ (*Figure 5A* and *Figure 5—figure supplement 2A to C*). The head groups of both lipids are bound around the middle of the membrane plane, with their acyl chains extending towards the rotor-stator interface. CDL1 is coordinated by R49 of ATPEG4, holding it near the loop connecting $H5_a$ and $H6_a$. Near the matrix half-channel, CDL2 is coordinated collectively by R160 of $H5_a$, R50 of ATPEG6 and R22 of subunit *k* (*Figure 5A*; *Figure 5—figure supplement 2B* to D). Together, these two bound cardiolipins enclose the two horizontal membrane helices $H5_a$ and $H6_a$, possibly acting to seal the $F_o$ against proton leakage by recruiting a high density of acyl chains, as well as separating lipid and aqueous environments in the vicinity of the two half-channels. Thus, in addition to previous studies suggesting transient interactions of the metazoan c-ring rotor with cardiolipin (*Duncan et al., 2016*), the *E. gracilis* ATP synthase structure shows that some of the cardiolipin is bound specifically to the stator, indicating a potential functional role in proton translocation.

## Peripheral $F_o$ subcomplex and lipid cavity

A cluster of phylum-specific subunits is located at the $F_o$ periphery, away from the dimer interface. Here, seven tightly associated subunits, ATPTB1, 6, 12 and ATPEG3, 5, 6, 8 form a subcomplex, connected to the conserved core in both, the lumen and matrix (*Figure 6A,B*). On the lumenal side, it is flanked by the terminal extensions of subunit *a* and subunit *k* (*Figure 6—figure supplement 1A,B*). The N-terminal ß-strand of subunit *a*, which is involved in the dimerization in yeast, shares a ß-sheet with ATPEG3, whereas the C-terminal of subunit *k* extending along the membrane plane interacts with six subunits of the subcomplex (all except ATPTB1). On the matrix side, ATPTB1 anchors the subcomplex to the conserved core through multiple contact sites. While the two moieties are extensively associated with each other outside the membrane, they are separated by more than 20-Å gap inside the membrane. Thus, a protein-enclosed membrane cavity is formed. In the cavity, we identified six bound cardiolipins (*Figure 6B*). To assess the lipid-binding capacity of the cavity, we performed coarse-grained molecular dynamics simulations of the entire mitochondrial ATP synthase dimer embedded in a phospholipid membrane containing 20% cardiolipin (*Figure 6—figure supplement 2*). The simulations indicate that the cavity is filled with a bilayer lipid array in which the lipid molecules can freely diffuse in or out of within the membrane (*Video 3*). Starting with random initial placement of phospholipids, the average residence time of cardiolipin was ~2.5 times higher than that of other lipid types included in the simulations (*Figure 6—figure supplement 2E*). The cardiolipin binding is induced by positively charged residues of subunits *a, i/j, k*, ATPTB1, ATPTB6, ATPEG3 extending into the cavity. To probe the dynamics of lipid binding, we calculated the probabilities of entering and remaining in the cavity. The lipid exhibiting the highest probability, especially over longer time intervals, is cardiolipin (*Figure 6—figure supplement 2C,D*), which is consistent with the assignment in the cryo-EM density maps (*Figure 6B*).

The membrane region around the identified $F_o$ cavity is displaced ~20 Å towards the lumen, thus contributing to its curvature (*Figure 6A* and *Video 1*). The displacement appears to be induced by

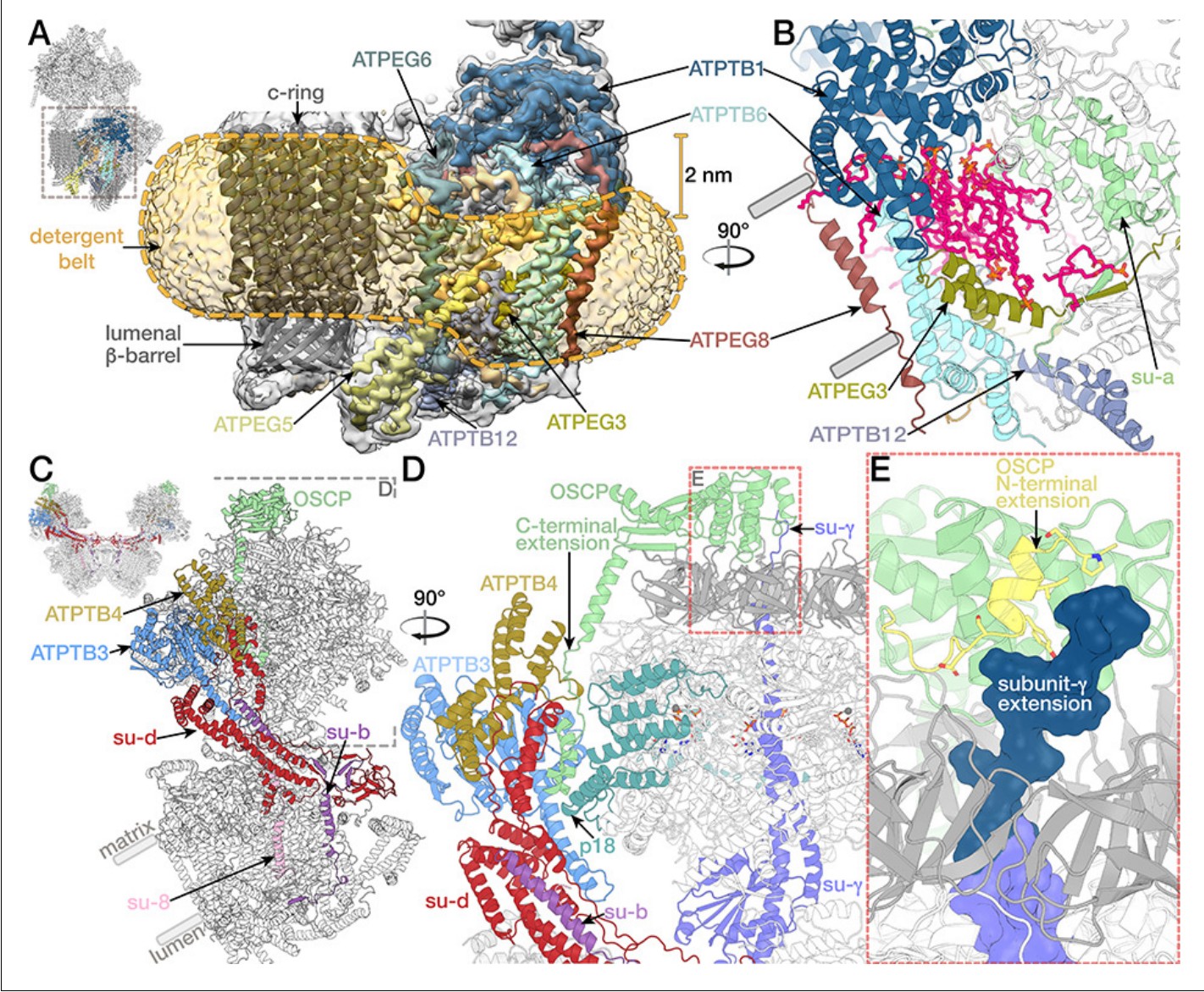

**Figure 6.** Peripheral F$_o$ subcomplex and the peripheral stalk. (**A**) Euglenozoa-specific subunits form a peripheral F$_o$ subcomplex. Density of the F$_o$ with proteins of the peripheral region coloured, c-ring model shown in grey, outline of the detergent belt (yellow dashed lines) with 2 nm offset towards the lumen indicated as determined by the density (transparent gold) (**B**) Atomic model of the F$_o$ periphery, cavity lipids are shown in magenta. (**C** to **E**) Attachment of the peripheral stalk to F$_1$. (**C**) *E. gracilis* ATP synthase with proteins constituting the peripheral stalk coloured. (**D**) Side view of the peripheral stalk tip and F$_1$ (white, crown domain light grey). C-terminal helix of OSCP (light green) extending towards the membrane attaches OSCP to the rest of the peripheral stalk via subunit *d* (red). (**E**) The N-terminal extension of OSCP (yellow) interacts with the C-terminal extension of the rotor subunit γ (conserved region purple, extension dark blue).

The online version of this article includes the following figure supplement(s) for figure 6:

**Figure supplement 1.** The peripheral F$_o$ subcomplex and inverted topology structural motif.

**Figure supplement 2.** Coarse-grained molecular dynamics simulations of the *E. gracilis* ATP synthase dimer.

**Figure supplement 3.** Interactions of OSCP extension with F$_1$ and the peripheral stalk.

two helical hairpin motifs of ATPTB1 and ATPEG3 extending into the membrane from opposite sides. This structural feature occludes the lipid-filled cavity in the membrane (*Figure 6—figure supplement 1A,B*). The architecture resembles that of an inverted-topology structural repeat found in excitatory amino acid transporters (*Figure 6—figure supplement 1C,D*), where similar helical hairpins play the key role in the transport mechanism and ion binding (*Crisman et al., 2009*). Within the

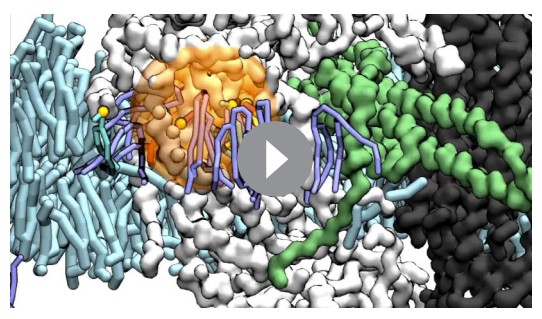

**Video 3.** Coarse-grained molecular dynamics simulation showing the diffusion of lipids into and out of the peripheral F$_o$ cavity (orange sphere, as described in *Figure 6—figure supplement 2C*) within the lipid bilayer over a period of 4 µs. A sliced view of the membrane bilayer is initially shown for reference, but later removed to allow viewing of the binding and unbinding phospholipids. Cardiolipin is indicated with purple, phosphatidic acid with yellow, phophatidyletanolamine with red, phosphatidylcholine with cyan acyl chains respectively. Lipids considered to be bound in the beginning or end of the simulation are visualized, demonstrating that cardiolipin replaces other lipids in the cavity during the simulation.

https://elifesciences.org/articles/51179#video3

membrane, the distances between the hairpin loops of a glutamate transporter homolog (*Yernool et al., 2004*) and ATPTB1/ATPEG3 are remarkably similar, within a 3.0–4.3 Å range. However, unlike in the intra-protein helical hairpin repeats, the membrane-spanning ATPTB1-ATPEG3 motif is formed by two separate subunits, and involves hydrophobic interactions (Y100$_{TB1}$, W55$_{EG3}$), which would restrict conformational variation.

## Peripheral stalk

The peripheral stalk connects the static membrane-embedded part to the catalytic subunits in the F$_1$, which in Euglenozoa is elaborated by three copies of subunit p18, each bound to subunit $\alpha$ (*Gahura et al., 2018*; *Montgomery et al., 2018*). The peripheral stalk in yeasts and mammals is built around a ~ 150 Å-long helix of subunit *b*, extending from the membrane to the functionally important and conserved oligomycin-sensitivity conferring protein (OSCP) on top of the F$_1$ crown (*Rees et al., 2009*). By contrast, subunit *b* is significantly shorter in *E. gracilis*, extending only ~65 Å into the matrix, and does not interact with OSCP. This is compensated by extensions of OSCP and subunit *d* mediating the attachment of the peripheral stalk, which is augmented by two species-specific soluble subunits ATPTB3 and ATPTB4 (*Figure 6C,D*; *Figure 6—figure supplement 3*). The additional elements significantly increase the size of the peripheral stalk compared to yeast. The C-terminal OSCP extension contains a 13-residue proline-rich region, followed by 26 residues arranged in two helices (D240-L246, V250-A265), which contact the large matrix extension of subunit *d*. This interaction is further clamped by ATPTB3 and ATPTB4 of the peripheral stalk tip and p18 on F$_1$ (*Figure 6D*). Thus, the *E. gracilis* peripheral stalk displays a different mode of interaction with the OSCP compared to all other reported ATP synthase structures.

The OSCP N-terminal domain contains a structurally conserved bundle of four $\alpha$-helices, followed by the N-terminal extension with a terminal helix. The N-terminal helix of OSCP connects to the central stalk through an interaction with the C-terminal extension of subunit $\gamma$ of the central stalk (*Figure 6E*). We observe this contact in all three rotational states, which provides the first observation of an interaction between the peripheral and central stalk in the ATP synthase (*Figure 7—figure supplement 1A* to C). This raises the question how central stalk rotation proceeds despite the observed anchoring. Studies in *Escherichia coli* F$_1$-ATPase showed that artificial crosslinking of rotor subunit $\gamma$ with stator subunit $\alpha$ near the crown domain does not impair the hydrolysis activity nor the full rotation of subunit $\gamma$ (*Gumbiowski et al., 2001*; *Hilbers et al., 2013*). Therefore, the interaction observed between the *E. gracilis* OSCP and subunit $\gamma$ is consistent with rotational catalysis. However, a permanent interaction would require rotation around a pivot point, presumably the short, extended linker region (K296-G298) in subunit $\gamma$, which is less well resolved in the cryo-EM maps. The flexibility of the central stalk has previously been suggested to play an important role in the energy transmission in the ATP synthase (*Guo et al., 2019*; *Murphy et al., 2019*; *Wächter et al., 2011*). Thus, the anchoring of subunit $\gamma$ may affect the energy required for transition between the individual rotational substeps. The extensions of OSCP and subunit $\gamma$ are conserved in Kinetoplastida (*Figure 6—figure supplement 3F* to H), suggesting that the observed interaction is a structural feature of euglenozoan ATP synthases. Together, the matrix-exposed subunit $\gamma$/OSCP interactions and lumenal ATPEG1/c-ring contacts (*Figure 1D*) pin the F$_1$/c-ring-complex to the stator, likely providing additional stability.

## Unusual binding mode of IF$_1$

IF$_1$ is a natural inhibitor protein of the F$_1$-subcomplex (αβ)$_3$γδε that interacts with subunits α, β and γ upon pH increase (*Cabezon et al., 2000*; *Cabezón et al., 2001*). Conventionally, it binds to F$_1$ via the C-terminal helix that protrudes into the β$_{DP}$-α$_{DP}$ interface, followed by a short N-terminal helix, which extends into the central F$_1$ cavity and interacts with the α-helical coiled coil of subunit γ of the central stalk to stall its rotation (*Figure 7B*) (*Cabezón et al., 2003*; *Gledhill et al., 2007*). Mechanistically, the central stalk rotation of the mammalian mitochondrial F$_1$-ATPase is specifically inhibited by the N-terminal of IF$_1$ that enters through the open catalytic β$_E$-α$_E$ interface, and the subsequent 120°-rotation of the central stalk is coupled to a deeper insertion of the α-helix towards subunit γ (*Bason et al., 2014*). Importantly, the minimal inhibitory region of IF$_1$ (A14-K47, mammalian numbering) has been shown to include the N-terminal helix (*van Raaij et al., 1996*).

The *E. gracilis* F$_1$/cring maps of rotational states 1 and 2 showed strong density for IF$_1$, but weak density in rotational state 3. Subsequent focused classification of the IF$_1$ binding site suggested complete inhibitor binding for rotational states 1 and 2, whereas rotational state 3 displayed a mixed occupancy, with a minority (34%) of particles containing IF$_1$ and a majority (66%) not containing the inhibitor (*Figure 1—figure supplement 1D*). Unlike the bovine ATP synthase, which displays a nearly equal particle distribution among the three rotational states (*Zhou et al., 2015*), rotational state-3 (20%) is less populated than rotational states 1 (44%) and 2 (36%). Rather than reflecting dwell times during rotary catalysis, the distribution of rotational states is likely dictated by accessibility of the respective catalytic interfaces to IF$_1$ binding, which for rotational state-3 may be sterically hindered by the proximity to the peripheral stalk.

In all three rotational states, IF$_1$ is bound to the β$_{DP}$-α$_{DP}$ interface adopting a similar conformation (*Figure 7—figure supplement 1A,B,D,E*, *Figure 1—figure supplement 1D*). While *E. gracilis* IF$_1$ shares the conserved C-terminal helix at the β$_{DP}$-α$_{DP}$ interface, including an EERY consensus motif, it does not contain the N-terminal helix. Instead, a Euglenozoa-specific proline residue (P46) follows the EERY motif and produces a break in the C-terminal helix of IF$_1$, which causes the N-terminal region to extend in the opposite direction compared to bovine IF$_1$. This results in interactions with the C-terminal domain of the α$_{DP}$-subunit via a number of residues that are conserved in Euglenozoa (*Figure 7D,E*). As a consequence, the *E. gracilis* IF$_1$, extends around the C-terminal domain of the α$_{DP}$-subunit and exits the F$_1$ through the β$_{TP}$-α$_{DP}$ interface in close proximity of the γ-protrusion of the central stalk (*Figure 7A*). Consequently, the N-terminus of the *E. gracilis* IF$_1$ does not contact the central α-helical coiled coil of the central stalk, indicating a different binding mode compared to the mammalian inhibitor. Importantly, the residues that anchor the extended N-terminal region of the *E. gracilis* IF$_1$ to the α$_{DP}$-subunit are conserved in Trypanosomatids, indicating that the newly identified binding mode for the inhibition of the mitochondrial ATP synthase is synthase is structurally conserved in the related group of parasites (*Figure 7—figure supplement 1E*).

The C-terminal region of both the bovine and *T. brucei* IF$_1$ has been characterised as a homo-oligomerisation domain, mediating the assembly of IF$_1$ into a dimer, which is the inhibitory form in both organisms (*Cabezón et al., 2001*; *Gahura et al., 2018*). Recently, mammalian IF$_1$ was shown to bridge neighboring mitochondrial ATP synthase dimers into a tetramer (*Gu et al., 2019*). Our unsharpened map of the ATP synthase dimer with both monomers in rotational state-1 (IF$_1$pointing towards the C$_2$-axis), shows a continuous density from the C-terminus of IF$_1$. This density extends towards the dimer interface and contacts the all-β-fold formed by subunits *b* and *d* (*Figure 7—figure supplement 1F,G*), suggesting that also in Euglenozoa both F$_1$ complexes in the ATP synthase dimer may be inhibited by a dimeric IF$_1$. However, in contrast to the mammalian ATP synthase, the specific binding mode of the Euglenozoan IF$_1$ assembly links two monomers within the dimer, rather than between neighbouring dimers.

## Conclusions

This work describes the composition and structure of a divergent mitochondrial ATP synthase of Euglenozoa with native lipids. The distinct subunit *a*, which had not been previously identified by genomic sequencing, is assigned directly from the density map, and markedly deviates from the structures in all the previously characterized ATP synthases. The newly found elements are involved in proton transfer, including functional substitution of the otherwise mitochondrially conserved glutamate of H6*a* with a histidine residue of H5*a*. The proton release channel is lined by species-specific

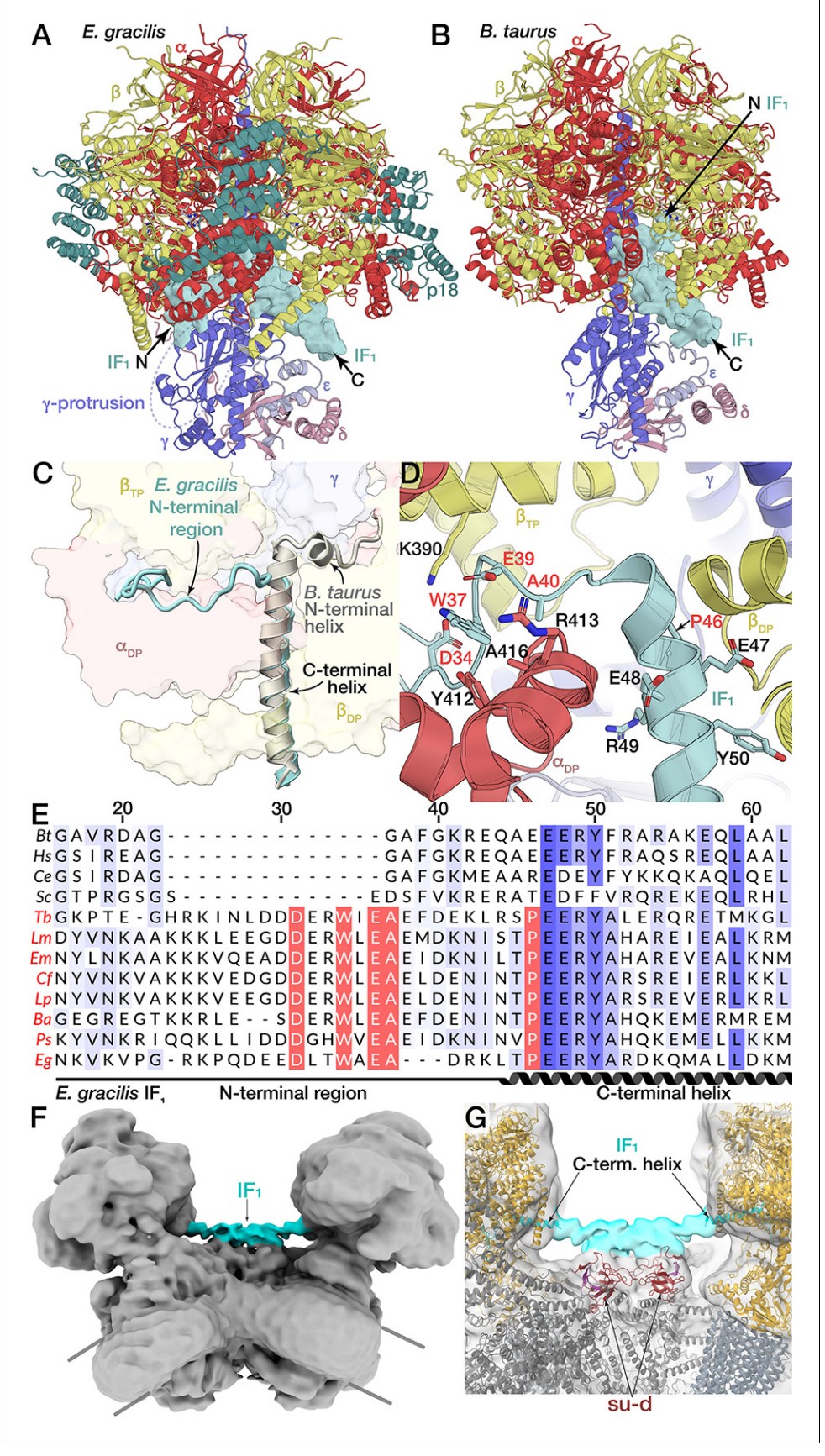

**Figure 7.** Unusual binding mode of IF$_1$. (**A** and **B**) Comparison of IF$_1$ binding mode between *E. gracilis* (rotational state-1) and *Bos taurus* (PDB ID: 2V7Q) (*Gledhill et al., 2007*). In the *E. gracilis* structure, both termini are located outside the F$_1$. (**C**) Superposition of the IF$_1$-inhibited *E. gracilis* F$_1$ and the bovine counterpart. Both structures share the conserved C-terminal helix but differ in the structure of the N-termini. (**D**) Close-up of the IF$_1$ binding site. The C-terminal helix of the *E. gracilis* IF$_1$ contains the conserved EERY, followed by the helix-breaking P46. In the Euglenozoa-specific N-terminal region W37 and E39 interact with $\alpha_{DP}$, whereas D34 interacts with $\beta_{TP.}$ (**E**) Multiple sequence alignment of IF$_1$ from different species; *Bos tarus* (Bt), *Homo sapiens* (Hs), *Caenorhabditis elegans* (Ce), *Saccharomyces cerevisiae* (Sc), the trypanosomatids *Trypanosoma brucei* (Tb), *Leishmania major* (Lm), *Endotrypanum monterogeii* (Em), *Crithidia fasciculata* (Cf), *Leptomonas pyrrhocoris* (Lp), *Blechomonas ayalai* (Ba), *Phytomonas sp.* (Ps) and the euglenoid *Euglena gracilis* (Eg). Euglenozoan species names shown in red. Residues are shaded in blue according to conservation. Residues conserved within Euglenozoa are highlighted red. (**F,G**) Bridging density in ATP synthase dimer. (**F**) *E. gracilis* ATP synthase dimer with both monomers in rotational state-1; map shown at low threshold. (**G**) From the C-terminal helix of IF$_1$ a continuous density (cyan) extends towards the C$_2$-symmetry axis, contacting the all-β fold formed by subunits *d* and *b*, thereby bridging the two F$_1$ subcomplexes.

The online version of this article includes the following figure supplement(s) for figure 7:

**Figure supplement 1.** Three main rotational states with bound nucleotides and IF$_1$.

---

subunits that together with four horizontal helices contribute to an almost double-sized contact area between the rotor and stator in the membrane. In addition, a direct interaction between the peripheral and central stalks is reported for the first time in the ATP synthase. Taken together, it implies a different energetic landscape for transition between the rotational substeps. This exemplifies the extreme structural divergence of the mechanistic components of ATP synthase.

The signature feature of mitochondrial ATP synthase dimers is the induction of membrane curvature that leads to cristae formation. In *E. gracilis* it is induced at the dimer interface, which is formed by species- or phylum-specific subunits and extensions of conserved subunits. Additionally, a membrane subcomplex is peripherally associated with conserved subunits, thereby contributing to a local displacement of the membrane towards the lumen. This suggests that interactions shaping the inner mitochondrial membrane are likely to have evolved independently. At the dimer interface, bound cardiolipins are specifically coordinated by different subunits, and the cavity formed by the newly identified subcomplex is predominantly occupied by cardiolipins. Therefore, protein-lipid interactions also play an important role in membrane bending. These observations extend our understanding of the structural mechanisms leading to a membrane curvature.

Last, the newly described binding mode of the inhibitor IF$_1$ to the ATP synthase is different from human, but appears to be conserved in Trypanosomatids, providing a potential therapeutic target.

## Materials and methods

### Cell culture and mitochondria isolation

*Euglena gracilis* was obtained from Lebendkulturen Helbig and grown in Hutner's modified media (*Buetow and Padilla, 1963*) in 2 l Erlenmeyer flasks (14 × 700 ml culture) at 25˚C and 120 rpm in the dark. Cells were harvested in late-logarithmic phase by centrifugation at 1200 xg, 10 min, 4˚C. The pelleted cells were transferred into a cold mortar and lysed by grinding them with 500-µm glass beads in buffer A (50 mM HEPES pH 7.5, 210 mM mannitol, 70 mM sucrose, 1 mM DTT, 1 mM EGTA, 5 mM EDTA) for 5 min. The lysate was centrifuged at 1500 xg, 10 min, 4˚C and the supernatant was spun again at 20,000 xg, 20 min, 4˚C. The resulting crude mitochondrial pellet was further purified on a discontinuous sucrose gradient in buffer B (20 mM HEPES-KOH pH 7.5, 2 mM EDTA, 15/23/32/60% sucrose) by centrifugation (103,745 xg, 1 hr, 4˚C) in an SW28 rotor (Beckman Coulter) and enriched mitochondria were collected from the 32–60% (w/v) interface.

### Isolation of *E. gracilis* ATP synthase dimers

Approximately 10 mg mitochondria were lysed in a total volume of 90 ml buffer C (25 mM HEPES/KOH pH 7.5, 25 mM KCl, 15 mM MgOAc$_2$, 1.7% Triton-X100, 2 mM DTT, one tablet EDTA-free Protease Inhibitor Cocktail) for 2 hr at 4˚C and the lysate was cleared by centrifugation at 30,000 xg, 20 min, 4˚C. The supernatant was layered on a sucrose cushion in buffer D (1 M sucrose, 25 mM

HEPES/KOH pH 7.5, 25 mM KCl, 15 mM MgOAc$_2$, 1% Triton-X100, 2 mM DTT) and centrifuged 158,420 xg, 3 hr, 4°C in a Ti45 rotor (Beckman Coulter). The resulting pellet was resuspended in 200 μl buffer E (25 mM HEPES/KOH pH 7.5, 25 mM KCl, 15 mM MgOAc$_2$, 2 mM DTT, 0.05% β-DDM) and gel filtrated over a Superose 6 Increase 3.2/300 column (GE Healthcare) in buffer E. Fractions corresponding to ATP synthase dimers were pooled and concentrated to 25 μl in a vivaspin500 filter (100 kDa MWCO).

## Electron cryo-microscopy and data processing

3 μl sample (~5 mg/ml) were applied to glow-discharged Quantifoil R1.2/1.3 Cu grids and vitrified by plunge-freezing into liquid ethane after blotting for 3 s. Samples were imaged on a Titan Krios operated at 300 kV at a magnification of 130 kx (1.05 Å/pixel) with a Quantum K2 camera (slit width 20 eV) at an exposure rate of 4 electrons/pixel/s with a 10-s exposure and 25 frames using the EPU software. A total of 9045 collected movies were motion-corrected and exposure-weighted using MotionCor2 (*Zheng et al., 2017*) and contrast transfer function (CTF) estimation was performed using Gctf (*Zhang, 2016*). All subsequent image processing was performed in RELION-3 (*Zivanov et al., 2018*). Bad images were removed manually by inspection in real and Fourier space. Initial rounds of Gaussian-based particle picking, classification and refinement were used to generate picking references from the data. Reference-based particle picking was performed using Gautomatch (developed by Dr Kai Zhang, MRC Laboratory of Molecular Biology, Cambridge, UK, http://www.mrc-lmb.cam.ac.uk/kzhang/Gautomatch) to pick 555,269 particles, which were subjected to reference-free two-dimensional (2D) classification, resulting in 540,669 particles for three-dimensional (3D) classification, from which 2 classes containing 171,033 particles were selected for a consensus 3D-refinement, applying C$_2$-symmetry. The resulting pre-aligned particles were C$_2$-symmetry expanded and a mask including one asymmetric unit was used for signal subtraction on the particles. Subsequent local-angular-search 3D-classification into three classes yielded three maps corresponding to the three main rotational states (*Figure 1—figure supplement 1D*). The class corresponding to rotational state-3 could be improved by sub-classification. Masked refinements of the F$_1$/c-ring regions yielded three final maps of rotational state-1 (150,744 particles), state-2 (122,085 particles) and state-3 (43,232 particles) at 3.04 Å, 3.14 Å and 3.92 Å resolution, respectively (*Figure 1—figure supplement 2C* to E). Rotational states 1 and 2 show strong density for IF$_1$, which is weak in rotational state-3. Focussed classification of the IF$_1$ binding site for rotational states 1 and 2 did not yield a class lacking IF$_1$, indicating complete occupancy. By contrast, rotational state-3 displayed a mixed occupancy, with a minority (34%) of particles containing IF$_1$ (refined to 4.2 Å resolution) and a majority (66%) not containing the inhibitor (refined to 4.1 Å resolution; *Figure 1—figure supplement 1D*). Masked refinements of the rotor (central stalk and c-ring) and peripheral stalk tip region in rotational state-1 yielded maps at 3.3 Å and 3.82 Å, respectively (*Figure 1—figure supplement 2F,G*). For the refinement of these maps, signal-subtracted particles were used for refinement, whereas original particles were used for final reconstruction. Masked refinement of combined particles from all three rotational states, yielded a 2.82 Å map of the static membrane region (*Figure 1—figure supplement 2A*). Finally, a consensus 4.32 Å resolution dimer map was generated by combining particles in which both F$_1$/c-ring subcomplexes are in rotational state-1 (*Figure 1—figure supplement 2B*). All final maps were generated from CTF-refined particles. All resolution estimates are according to Fourier shell correlations (FSC) that were calculated from independently refined half-maps using the 0.143-criterion with correction for the effect of the applied masks (*Figure 1—figure supplement 1C*).

## Atomic model building and refinement

Atomic model building was performed in *Coot* (*Emsley and Cowtan, 2004*). F$_o$ subunits were built de novo and identified from the density map as previously reported *E. gracilis* ATP synthase subunits (*Perez et al., 2014*; *Yadav et al., 2017*) or newly identified subunits using the *E. gracilis* transcriptome or genome (NCBI accession code: PRJEB27422). OSCP/F$_1$/c-ring models were built using a homology model (*Waterhouse et al., 2018*) of the yeast F$_1$/c$_{10}$-ring (PDB ID: 3ZRY) (*Giraud et al., 2012*), whereas OSCP, IF$_1$ and p18 were built de novo. Real-space refinement of atomic models was performed in PHENIX (*Afonine et al., 2018*) using secondary structure restraints. Cardiolipins resolved in the cryo-EM map were identified by their unique structure containing two phosphatidyl

groups linked by a central glycerol bridge (*Video 2*). Bound lipids other than cardiolipin could not be unambiguously identified from the head group of their phosphatidyl density and were thus modelled as phosphatidic acid. Acyl tails of lipids were truncated according to map density. The atomic model of the rotor in rotational state-1 was rigid-body fitted into the local-resolution-filtered $F_1$/c-ring maps of all three rotational states and the combined $F_1$/c-ring models were subsequently refined with reference restraints on the c-ring. To generate a composite model of the complete ATP synthase dimer, the atomic models of the membrane region, the OSCP/$F_1$/c-ring in rotational-state one and the peripheral stalk tip were combined after rigid-body fitting into the consensus map of the dimer with both rotors in rotational state-1 and refined in PHENIX using reference restraints. Model statistics were calculated using MolProbity (*Chen et al., 2010*). To evaluate potential overfitting of the atomic model during refinement, the atomic coordinates of the refined model were randomly displaced by shifts of up to 0.5 Å using 'Shake' in the CCPEM suite (*Burnley et al., 2017*). The shaken model was Real-space refined using PHENIX against one half map that had been reference-sharpened using Refmac (*Brown et al., 2015*). Subsequently, $FSC_{work}$ and $FSC_{test}$ between the model and the two unfiltered half-maps, were calculated as described (*Brown et al., 2015*) (*Figure 1—figure supplement 2*).

## Data analysis and visualisation

The lumenal half-channel was traced as a void in the $F_o$-model, using HOLLOW (*Ho and Gruswitz, 2008*). The matrix half channel was visualised as a void (inverted contrast) in the density map of the membrane region. Images were rendered using PyMOL 2 (Schrödinger, LLC), Chimera (*Pettersen et al., 2004*) or ChimeraX (*Goddard et al., 2018*). Sequence alignments were performed using MAFFT (*Katoh et al., 2002*). Prediction of cleavage sites of the mitochondrial matrix protease was performed using MitoFates (*Fukasawa et al., 2015*). To assess the presence of the newly identified subunit *a* and subunit 8 in the *E. gracilis* mitochondrial genome, a BLAST search of the DNA sequence (experiment PRJNA294935; *Dobáková et al., 2015*) was performed, which showed that the query is entirely covered by individual sequencing reads. The respective sequences were subsequently used to assemble a consensus contig using CAP3 (*Huang and Madan, 1999*) (*Figure 2—figure supplement 1E*).

## Molecular dynamics simulations

The consensus ATP synthase dimer model was placed in a planar bilayer (45% palmitoyl-oleoyl-phosphatidylcholine (POPC), 30% palmitoyl-oleoyl-phosphatidylethanolamine (POPE), 20% cardiolipin (CDL), 5% palmitoyl-oleoyl-phosphatidic acid (POPA); corresponding to 3609, 3011, 1604, and 400 molecules, respectively) using *insane* (*Wassenaar et al., 2015*), surrounded by an aqueous 150 mM NaCl solution. Three replicas with identical lipid types and quantities were generated, differing only in the initial lipid placement. The protein model was converted to a coarse-grained MARTINI representation using the *martinize* script (*de Jong et al., 2013*). Parameters for the lipids POPC, POPE, POPA, and CDL were used from the MARTINI force field for lipids, and parameters for the coarse-grained protein, water and ions were used from the MARTINI 2.0 forcefield (*Marrink et al., 2007*; *Monticelli et al., 2008*). The initial system box size was 50 × 50×35 nm with a total number of ~775,000 beads. Equilibration of the system was carried out by an energy minimization using the steep integrator for 10 ps total, followed by seven steps during which positional restraints on the protein and lipid beads were gradually lowered, while increasing the time step of the simulation, until a total of 91 ns equilibration time. The Berendsen semi-isotropic pressure coupling was used during equilibration (*Berendsen et al., 1984*). During production, three replicas of 4 µs each were simulated in 20-fs time steps using periodic boundary conditions. The protein bead scaffold was restrained utilizing the EleNeDyn forcefield (*Periole et al., 2009*) with a force constant of 500 kJ/mol/nm$^2$, and all beads were held with a positional restraint constant of 10 kJ/mol/nm$^2$. The temperature of the system was maintained at 298 K using the velocity-rescaling thermostat (*Bussi et al., 2007*) with temperature coupling separately for protein, lipids, and solvent. The pressure was maintained at 1 bar, using the Parrinello-Rahman barostat and semi-isotropic coupling (*Parrinello and Rahman, 1981*). Electrostatics were calculated with a dielectric constant of 15.0 and a cutoff of 1.1 nm was applied for short-range interactions. Molecular dynamics simulations were performed with GROMACS 2016.1 (*Abraham et al., 2015*).

Simulation data was visualised using VMD or the python package matplotlib (*Humphrey et al., 1996*; *Hunter, 2007*). The script *cg_bonds* (available on cgmartini.nl) was used to visualize the coarse-grained beads as whole molecules. The radial distribution function, as well as the distances between the $F_o$ peripheral cavity and the lipid molecules were determined using GROMACS tools *rdf* and *pairdist* (*Abraham et al., 2015*), respectively. Residence time in the pocket was smoothed to avoid intermittent exit of lipids from the $F_o$ peripheral cavity. A lipid was considered to begin a binding event if any of its beads reached within a 1.8 nm distance from the center of mass of the cavity, and was considered to be bound until leaving a buffer zone of 3.5 nm distance from the cavity center of mass. Error bars in *Figure 6—figure supplement 1E* represent a confidence interval of 90%, obtained after bootstrapping of the original distribution. The nonparametric Mann-Whitney U test was used to determine if the residence times of each lipid type originated from the same distribution as cardiolipin residence times (*Mann and Whitney, 1947*).

Over different time spans ($\tau$), the probability of different lipid types (j) to enter, remain in, or exit the binding site was calculated. For each lipid type, the number of times ($N_j$) any bead of each lipid molecule of this type entered the binding site ($N_j(o_k \rightarrow i_{k+\tau})$), or remained in the binding site ($N_j(i_k \rightarrow i_{k+\tau})$) where k is the frame index in the simulation, was calculated. This quantity was normalised over the total number of possible events, including the other two possible binding events, a lipid molecule exiting the binding site ($N_j(i_k \rightarrow o_{k+\tau})$), and a lipid molecule remaining outside the binding site ($N_j(o_k \rightarrow o_{k+\tau})$), thus yielding a probability. Time spans $\tau$ of 1 ns to 3000 ns were considered. For example, the probability of entering the pocket ($N_j(o_k \rightarrow i_{k+\tau})$) is expressed as:

$$\rho(o \rightarrow i)_{\tau, j} = \frac{\sum_{j=1}^{N\ lipid\ types} \sum_{k=1}^{N\ frames} \left(N_j(o_k \rightarrow i_{k+\tau})\right)}{\sum_{j=1}^{N\ lipid\ types} \sum_{k=1}^{N\ frames} \left(N_j(o_k \rightarrow i_{k+\tau})\right) + \left(N_j(i_k \rightarrow i_{k+\tau})\right) + \left(N_j(i_k \rightarrow o_{k+\tau})\right) + \left(N_j(o_k \rightarrow o_{k+\tau})\right)}$$

## Acknowledgements

The data was collected at the SciLifeLab cryo-EM facility funded by the Knut and Alice Wallenberg, Family Erling Persson, and Kempe foundations. We thank M Field and T Ebenezer for providing to unpublished *E. gracilis* sequence data, the members of the Amunts lab for active discussions throughout the project, and L Delemotte for contributing to the data analysis. This work was supported by the Swedish Foundation for Strategic Research (FFL15:0325), Ragnar Söderberg Foundation (M44/16), Swedish Research Council (NT_2015–04107), Cancerfonden (2017/1041), European Research Council (ERC-2018-StG-805230), Knut and Alice Wallenberg Foundation (2018.0080) and an European Molecular Biology Organization Long-Term Fellowship (ALTF 260–2017 to AM). AA is supported by the European Molecular Biology Organization Young Investigator Program.

## Additional information

### Funding

| Funder | Grant reference number | Author |
| --- | --- | --- |
| Stiftelsen för Strategisk Forskning | FFL15:0325 | Alexey Amunts |
| Ragnar Söderbergs stiftelse | M44/16 | Alexey Amunts |
| Vetenskapsrådet | NT_2015-04107 | Alexey Amunts |
| Cancerfonden | 2017/1041 | Alexey Amunts |
| H2020 European Research Council | ERC-2018-StG-805230 | Alexey Amunts |
| Knut och Alice Wallenbergs Stiftelse | 2018.0080 | Alexey Amunts |
| European Molecular Biology Organization | ALTF 260-2017 | Alexander Mühleip |
| European Molecular Biology Organization | Young Investigator Program | Alexey Amunts |

The funders had no role in study design, data collection and interpretation, or the decision to submit the work for publication.

## Author contributions

Alexander Mühleip, Conceptualization, Data curation, Formal analysis, Funding acquisition, Validation, Investigation, Methodology, Writing—original draft, Writing—review and editing; Sarah E McComas, Data curation, Formal analysis, Investigation; Alexey Amunts, Resources, Formal analysis, Supervision, Funding acquisition, Investigation, Writing—original draft, Project administration, Writing—review and editing

## Author ORCIDs

Alexander Mühleip  https://orcid.org/0000-0002-1877-2282
Alexey Amunts  https://orcid.org/0000-0002-5302-1740

## Decision letter and Author response

Decision letter https://doi.org/10.7554/eLife.51179.sa1
Author response https://doi.org/10.7554/eLife.51179.sa2

# Additional files

## Supplementary files

• Supplementary file 1. Cryo-EM data collection.

• Supplementary file 2. *E. gracilis* ATP synthase dimer atomic model statistics *FSC corrected for the effect of the mask according to 0.143-cutoff criterion ** FSC (masked) according to 0.5-cutoff criterion.

• Supplementary file 3. Subunits of the *E. gracilis* ATP synthase dimer identified in this study.

• Transparent reporting form

## Data availability

All data generated or analyzed during this study are included in this Article and the Supplementary Information. The cryo-EM maps have been deposited in the Electron Microscopy Data Bank with accession codes EMD-10467, EMD-10468, EMD-10469, EMD-10470, EMD 10471, EMD-10472, EMD-10473, EMD-10474, EMD-10475. The atomic models have been deposited in the Protein Data Bank under accession codes 6TDU, 6TDV, 6TDW, 6TDX, 6TDY, 6TDZ, 6TE0.

The following datasets were generated:

| Author(s) | Year | Dataset title | Dataset URL | Database and Identifier |
|---|---|---|---|---|
| Mühleip A, McComas SE, Amunts A | 2019 | Atomic Models | http://www.rcsb.org/structure/6TDU | Protein Data Bank, 6TDU |
| Mühleip A, McComas SE, Amunts A | 2019 | Atomic Models | http://www.rcsb.org/structure/6TDV | Protein Data Bank, 6TDV |
| Mühleip A, McComas SE, Amunts A | 2019 | Atomic Models | http://www.rcsb.org/structure/6TDW | Protein Data Bank, 6TDW |
| Mühleip A, McComas SE, Amunts A | 2019 | Atomic Models | http://www.rcsb.org/structure/6TDX | Protein Data Bank, 6TDX |
| Mühleip A, McComas SE, Amunts A | 2019 | Atomic Models | http://www.rcsb.org/structure/6TDY | Protein Data Bank, 6TDY |
| Mühleip A, McComas SE, Amunts A | 2019 | Atomic Models | http://www.rcsb.org/structure/6TDZ | Protein Data Bank, 6TDZ |
| Mühleip A, McComas SE, Amunts A | 2019 | Atomic Models | http://www.rcsb.org/structure/6TE0 | Protein Data Bank, 6TE0 |
| Mühleip A, McComas SE, Amunts A | 2019 | Cryo-EM maps | https://www.ebi.ac.uk/pdbe/entry/emdb/EMD-10467 | Electron Microscopy Data Bank, EMD-10467 |

| | | | | |
|---|---|---|---|---|
| Mühleip A, McComas SE, Amunts A | 2019 | Cryo-EM maps | https://www.ebi.ac.uk/pdbe/entry/emdb/EMD-10468 | Electron Microscopy Data Bank, EMD-10468 |
| Mühleip A, McComas SE, Amunts A | 2019 | Cryo-EM maps | https://www.ebi.ac.uk/pdbe/entry/emdb/EMD-10469 | Electron Microscopy Data Bank, EMD-10469 |
| Mühleip A, McComas SE, Amunts A | 2019 | Cryo-EM maps | https://www.ebi.ac.uk/pdbe/entry/emdb/EMD-10470 | Electron Microscopy Data Bank, EMD-10470 |
| Mühleip A, McComas SE, Amunts A | 2019 | Cryo-EM maps | https://www.ebi.ac.uk/pdbe/entry/emdb/EMD-10471 | Electron Microscopy Data Bank, EMD 10471 |
| Mühleip A, McComas SE, Amunts A | 2019 | Cryo-EM maps | https://www.ebi.ac.uk/pdbe/entry/emdb/EMD-10472 | Electron Microscopy Data Bank, EMD-10472 |
| Mühleip A, McComas SE, Amunts A | 2019 | Cryo-EM maps | https://www.ebi.ac.uk/pdbe/entry/emdb/EMD-10473 | Electron Microscopy Data Bank, EMD-10473 |
| Mühleip A, McComas SE, Amunts A | 2019 | Cryo-EM maps | https://www.ebi.ac.uk/pdbe/entry/emdb/EMD-10474 | Electron Microscopy Data Bank, EMD-10474 |
| Mühleip A, McComas SE, Amunts A | 2019 | Cryo-EM maps | https://www.ebi.ac.uk/pdbe/entry/emdb/EMD-10475 | Electron Microscopy Data Bank, EMD-10475 |

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
