## [Decision Letter]

**Acceptance summary:**

The manuscript by Muhlheip et al. is a beautifully executed piece of work, presenting a cryo-EM structure of the F-ATPase of a Type IV mitochondrial ATP synthases from *Euglena gracilis*. The parasitic nature of this complex gives it an added interest especially in the architecture of subunit *a*.

**Decision letter after peer review:**

Thank you for submitting your article "Structure of a mitochondrial ATP synthase with native boundary cardiolipins" for consideration by *eLife*. Your article has been reviewed by three peer reviewers, one of whom is a member of our Board of Reviewing Editors, and the evaluation has been overseen by Cynthia Wolberger as the Senior Editor. The following individual involved in review of your submission has agreed to reveal their identity: John L Rubinstein (Reviewer #2).

The reviewers have discussed the reviews with one another and the Reviewing Editor has drafted this decision to help you prepare a revised submission.

Essential revisions:

1) The manuscript should be revised to discuss the previous literature more widely. At the moment some of the claims of novelty come across as being over sold. For example:

a) It seems quite well established that different species of mitochondrial ATP synthases have "dimerization interfaces made up of different subunits with no apparent homology" (Kühlbrandt, 2019). Rather than selling this as one of your major discoveries you should explain how your structure illustrates this principle (in in exquisite detail)

b) Cardiolipins are well known to be important for ATP synthase activity and this paper identifies lots of them. Contrary to the author's claims some have been observed at high resolution before (Klusch, 2017) at what I think is also the rotor/stator interface. This paper needs to be cited and discussed.

2) A cryo-EM data collection table is missing, including parameters such as defocus range that are otherwise absent from the manuscript. The authors must include this.

3) In my opinion, the biggest omission from the manuscript is a fulsome description of the interactions between ATPEG1 and the β-barrel formed by the N-terminal extensions from the c-ring. This remarkable feature suggests some sort of communication of information from the c-ring to the rest of the F_o_ region or vice versa. Further, it resembles a feature seen in the bovine mitochondrial ATP synthase monomer (Zhou et al., 2015) where subunit *e* curves back to contact the c-ring, while in yeast the subunit *e* takes on a different orientation. The authors should describe this interaction in more detail.

4) I was less convinced by the argument on the bound lipids from the data shown. The presence of lipids is not controversial but it is hard to gauge the reliability of the fitting from the data shown. For the bound lipids described, density is only shown for a handful and this is not unambiguous, for example in Figure 4G what is the extra unexplained density above the lipid? By not showing the surrounding map density it is hard to see the "noise" level and therefore attribute confidence in the docking. A supplementary figure to show the density for the lipids would help as would at least one example to show the density with respect to the surrounding density. It can be very noisy with the detergent micelle. It is understandable that the tail density is poorly resolved but it does make the accurate assignment hard. The MD simulation does not provide a convincing argument in the way it is shown. It shows that there is preference for CDL but the simulation could be better explained. In the supporting video the lipids appear to float around and come in and out of the bilayer with ease (although the bilayer is not shown). Is there a bilayer present that would hinder the lipids being removed from the cavity? It feels like this should be a 2D search within the bilayer to see which lipids have a preference for the binding site. The video shown makes it look like lipids are simply being added with no constraints on the difficulty of leaving the bilayer to float around in the intracellular space.

---

## [Author Response]

Essential revisions:1) The manuscript should be revised to discuss the previous literature more widely. At the moment some of the claims of novelty come across as being over sold. For example:a) It seems quite well established that different species of mitochondrial ATP synthases have "dimerization interfaces made up of different subunits with no apparent homology" (Kühlbrandt, 2019). Rather than selling this as one of your major discoveries you should explain how your structure illustrates this principle (in in exquisite detail)

To further expand on the description of the structural data, we added Figure 4—figure supplement 2, illustrating that the dimerization interface is composed not only of species/phylum-specific proteins (as suggested in the citation), but also of extensions of apparent homologs such as subunit *d* (previously annotated as phylum-specific ATPTB2) and subunit *f*.

As suggested, we deleted the sentence referring to the dimer interface being formed “exclusively by species-specific elements” from the conclusions and replaced it with “the dimer interface, which is formed by species- or phylum-specific subunits and extensions of conserved subunits”, thus explaining the mixed contributions to the dimerization interface, which is the new finding of this study.

b) Cardiolipins are well known to be important for ATP synthase activity and this paper identifies lots of them. Contrary to the author's claims some have been observed at high resolution before (Klusch, 2017) at what I think is also the rotor/stator interface. This paper needs to be cited and discussed.

Klusch et al., 2017, tentatively described a density feature in the 3.7 Å resolution *Polytomella* ATP synthase map as “consistent with a cardiolipin molecule”. However, no cardiolipin was modelled in the deposited structure (pdb code 6F36). More importantly, in the significantly improved 2.8 Å resolution structure (Murphy et al., 2019) no modeled cardiolipins have been reported. Instead all lipids are phosphatidylethanolamine (PEV, pdb code 6RD5), including the previously mentioned density region. Therefore there are no atomic model coordinates available for the suggested structural comparison and we would consider it misleading to discuss an obsolete density assignment.

In the revised manuscript, we now cite the previous observation of lipids in the F_o_ region; and clarify that previously reported atomic models do not contain cardiolipin (Introduction, second paragraph).

2) A cryo-EM data collection table is missing, including parameters such as defocus range that are otherwise absent from the manuscript. The authors must include this.

Supplementary file 1 has been added.

3) In my opinion, the biggest omission from the manuscript is a fulsome description of the interactions between ATPEG1 and the β-barrel formed by the N-terminal extensions from the c-ring. This remarkable feature suggests some sort of communication of information from the c-ring to the rest of the F_o_ region or vice versa. Further, it resembles a feature seen in the bovine mitochondrial ATP synthase monomer (Zhou et al., 2015) where subunit e curves back to contact the c-ring, while in yeast the subunit e takes on a different orientation. The authors should describe this interaction in more detail.

We thank the reviewers for giving us the opportunity to expand the discussion, which now appears in the last paragraph of the subsection “Identification of the transmembrane subunits” and in the second paragraph of the subsection “Unusual binding mode of IF_1_”. We cited the lumenal interaction observed in the bovine ATP synthase monomer (Zhou et al., 2015) and rephrased the mention of the proposed interaction in the porcine ATP synthase tetramer). The interacting residues have been indicated in Figure 1D.

4) I was less convinced by the argument on the bound lipids from the data shown. The presence of lipids is not controversial but it is hard to gauge the reliability of the fitting from the data shown. For the bound lipids described, density is only shown for a handful and this is not unambiguous, for example in Figure 4G what is the extra unexplained density above the lipid? By not showing the surrounding map density it is hard to see the "noise" level and therefore attribute confidence in the docking. A supplementary figure to show the density for the lipids would help as would at least one example to show the density with respect to the surrounding density. It can be very noisy with the detergent micelle. It is understandable that the tail density is poorly resolved but it does make the accurate assignment hard.

We thank the reviewer for drawing our attention to providing more experimental data on the bound lipids. The manuscript now includes Video 2, focusing on a cardiolipin with its cryo-EM map density with respect to the surrounding protein and detergent belt density that addresses the raised points. Regarding the ‘unexplained density above the lipid’ in Figure 4G, it is modeled as phosphatidic acid (PA) because monophosphatidyl lipids cannot be unambiguously distinguished by their head group densities (subsection “Overall structure”). Thus, “unexplained density” in the head group region is expected. To avoid misunderstanding, we have further clarified the criteria for lipid assignment in the Materials and methods section (subsection “Molecular dynamics simulations”).

The MD simulation does not provide a convincing argument in the way it is shown. It shows that there is preference for CDL but the simulation could be better explained. In the supporting video the lipids appear to float around and come in and out of the bilayer with ease (although the bilayer is not shown). Is there a bilayer present that would hinder the lipids being removed from the cavity? It feels like this should be a 2D search within the bilayer to see which lipids have a preference for the binding site. The video shown makes it look like lipids are simply being added with no constraints on the difficulty of leaving the bilayer to float around in the intracellular space.

We have clarified that all lipids are packed in a bilayer and are capable of diffusing freely across the bilayer, in or out of the protein cavities (subsection “Peripheral F_o_ subcomplex and lipid cavity”, first paragraph). Only selected relevant lipids from the bilayer are shown in Video 3 to visualize binding/unbinding events. To clarify further, we have referred to Figure 6—figure supplement 2C, indicating the position of the cavity in the protein, changed the viewing angle in the simulation video and included a cross-section of the bilayer in the beginning of the video.